

# Recombination between T-DNA insertions to cause chromosomal deletions in *Arabidopsis* is a rare phenomenon

John F. Seagrist[1,*], Shih-Heng Su[2,*] and Patrick J. Krysan[1,3]

[1] Department of Horticulture, University of Wisconsin-Madison, Madison, WI, United States of America
[2] Laboratory of Genetics, University of Wisconsin-Madison, Madison, WI, United States of America
[3] Genome Center of Wisconsin, University of Wisconsin-Madison, Madison, WI, United States of America
[*] These authors contributed equally to this work.

Corresponding author
Patrick J. Krysan, pjkrysan@wisc.edu

## ABSTRACT

We previously described the identification of a chromosomal deletion in *Arabidopsis thaliana* that resulted in the elimination of genomic DNA between two T-DNA insertions located ca. 25 kilobases apart on chromosome IV. The mechanism responsible for this deletion appears to have been recombination between the closely spaced T-DNA elements located in trans in a parent plant. In our original study, we observed one such deletion event after screening ca. 2,000 seedlings using a polymerase chain reaction (PCR) assay. Because a method for precisely deleting a selected region of the *Arabidopsis* genome would have significant value as a research tool, we were interested in determining the frequency with which this type of T-DNA-directed deletion occurs. To do this we designed a genetic screen that would allow us to phenotypically screen for deletions caused by recombination between T-DNA inserts. This screen involved crossing T-DNA single-mutant lines in order to produce F1 plants in which the two T-DNA insertions flanked a *FUSCA* (*FUS*) locus present in the genome. Loss-of-function mutations of *FUS* genes cause a distinctive developmental phenotype that can be easily scored visually in young seedlings. We used T-DNA lines flanking *FUS2*, *FUS6*, *FUS7*, and *FUS11* for this study. Recombination between the T-DNAs in an F1 parent should result in deletion of the *FUS* gene located between the T-DNAs. Because the deletion would be heterozygous in the F2 generation, we screened the F3 progeny of pooled F2 individuals to search for the *fus* loss-of-function phenotype. Using this strategy we were able to evaluate a total of 28,314 meioses for evidence of deletions caused by recombination between the T-DNA inserts. No seedlings displaying the *fus* phenotype were recovered, suggesting that deletions caused by recombination between T-DNA inserts are relatively rare events and may not be a useful tools for genome engineering in *Arabidopsis*.

## INTRODUCTION

In order to understand the function of a genome, it is critical to have tools available for modifying the DNA sequence of that genome. In the case of the model plant *Arabidopsis thaliana*, one of the most widely used strategies for knocking out a gene of interest is insertional mutagenesis based on the transferred DNA (T-DNA) of *Agrobacterium tumefaciens* (*Alonso et al., 2003*; *Krysan, Young & Sussman, 1999*; *O'Malley & Ecker, 2010*; *Parinov & Sundaresan, 2000*). A number of labs throughout the world have produced large populations of *Arabidopsis* transgenic lines in which T-DNA elements are randomly inserted throughout the genome (*Alonso et al., 2003*; *Brunaud et al., 2002*; *Ito et al., 2002*; *Rosso et al., 2003*; *Sessions et al., 2002*; *Woody et al., 2007*). The exact positions of the T-DNAs are determined by DNA sequencing (*Alonso et al., 2003*). Currently, there are over 400,000 T-DNA insertion lines cataloged in the *Arabidopsis* genome, with the corresponding seed available through public stock centers. Although the T-DNA elements are inserted "randomly", the existence of such a large catalog of lines means that most of the genes in *Arabidopsis* have at least one T-DNA mutant allele available. These public T-DNA collections are an extremely popular community resource for studying gene function in *Arabidopsis*. Because approximately 17% of the genes in *Arabidopsis* are members of a tandemly-duplicated gene family, however, it can be challenging to study gene function using T-DNA knockouts when the gene of interest has a second copy nearby on the same chromosome (*Initiative, 2000*). If these duplicated genes display functional redundancy, one must knockout both genes in order to reveal a mutant phenotype. Because of the extremely tight genetic linkage between tandemly duplicated genes, it is not practical to rely on meiotic recombination to produce a double-mutant for tandemly duplicated genes using T-DNA insertion lines.

The *Arabidopsis* MAP kinase kinase kinase gene *MEKK1* is part of a tandemly duplicated gene family in which *MEKK1*, *MEKK2*, and *MEKK3* are located within a ca. 25 kilobase (kb) region of chromosome IV. In a previous study we reported the identification of a deletion mutant that appeared to be the result of recombination between a T-DNA insertion in *MEKK1* and a second T-DNA located ca. 21 kb away in *MEKK3* (*Su et al., 2013*). Because the two T-DNA insertions were produced using the same vector, meiotic recombination directed by the homology within the T-DNA vector sequence seems to have been responsible for producing the deletion of the intervening genomic DNA. This deletion mutant proved to be valuable for dissecting the function of the *MEKK1/2/3* gene family (*Su et al., 2013*).

Due to the prevalence of tandemly-duplicated gene families in *Arabidopsis*, we were interested in determining if deletions caused by recombination between T-DNA inserts might be useful tools for performing genome engineering in *Arabidopsis*. In our study of the *MEKK1/2/3* tandem gene family, we identified the deletion mutant using a PCR-based screen of ca. 2,000 seedlings. Since we only found one individual carrying the deletion, it was not possible to accurately estimate the frequency with which this type of T-DNA directed deletion might occur. In order for deletions caused by recombination between T-DNA insertions to be useful genome engineering tools, the frequency with which it

occurs needs to be within a practical range to make screening for deletions cost- and time-effective. We therefore designed a genetic screen that enabled us to phenotypically screen for deletions caused by recombination between T-DNA inserts at a number of locations in the *Arabidopsis* genome. Despite screening 28,314 meioses for evidence of T-DNA-directed deletions, none were found. These results suggest that deletions caused by recombination between T-DNA inserts may be relatively rare events and that their utility for genome engineering may be limited.

## MATERIALS AND METHODS

### Plant material and growth conditions

*Arabidopsis* lines carrying T-DNA insertions flanking *FUS2* (At4g10180), FUS6 (At3g61140), *Fus7* (At4g14110), and *Fus11* (At5g14250) were obtained from the *Arabidopsis* Biological Resource Center at The Ohio State University (abrc.osu.edu). *FUS2*: SALK_046021 and SALK_004560. *FUS6*: WiscDsLox453-456F5, WiscDsLox402F03, SALK_203944, and SALK_062171. *FUS7*: SALK_073580 and SALK_150313. *FUS11*: SALK_150318 and SALK_150319. Seedlings were germinated on 1% agar (w/v) plates containing 0.5× Murashige and Skoog (MS) basal salt mixture under continuous light at 20 °C–23 °C. Seedlings were transferred to a 2:1 mixture (vol:vol) of potting mix:pearlite 5–7 days after germination and grown under constant light at 20 °C–23 °C.

### Genotyping

PCR-based genotyping was used to determine the presence of the desired T-DNA insertion elements. Genomic DNA was extracted from 5 mm square pieces of leaf tissue as previously described (*Kasajima et al., 2004*). PCR was performed using the gene-specific PCR primers indicated in Table 1 in conjunction with the T-DNA left border primer p745: 5′-AACGTCCGCAATGTGTTATTAAGTTGTC-3′. The thermal cycling conditions used were as follows: 96 °C for 2 min, followed by 40 cycles of 94 °C for 10 s, 58 °C for 30 s, and 72 °C for 30 s. PCR products were visualized by agarose gel electrophoresis followed by ethidium bromide staining.

### Phenotypic screening

F1 plants were produced by crossing two single-mutant T-DNA lines. PCR genotyping was used to confirm that the F1 plants carried the desired T-DNA insertions. F2 seed was then collected from individual F1 plants. F2 seedlings were germinated on petri dishes containing agar growth media and transplanted to potting mix at a density of 9 seedlings per 3.5 inch square pot. The F2 plants were allowed to self-pollinate, and the resulting seed was collected in bulk from each pot of 9 plants. Ca. 1,200 F3 seeds from each pool of nine F2 plants were placed onto water-saturated 125 mm diameter circular pieces of Whatman filter paper (Whatman cat. #1004-125) inside 140 mm diameter plastic petri dishes. The plates were then incubated in the dark at 4 °C for 2–3 days to stratify the seeds. The plates were then placed under light at 20 °C–23 °C for four hours to stimulate germination, and then transferred to black box with no light at 20 °C–23 °C for four days to allow for etiolated growth. Plates were then visually inspected to search for seedlings displaying the distinctive *fus* phenotype.
**Table 1  PCR primers.**

| Gene | T-DNA | Primer F | Primer R | T-DNA PCR[a] |
|---|---|---|---|---|
| FUS2 | SK_046021 | CCGATCAACAAAATGGGAAGT | TCACATTTTGGATCGGATTTT | F+p745 |
| | SK_004560 | TCCATTAACTCCAGATTTGTGAA | GTAATCGGACCAAAGTTACGG | F+p754 or R+p745 |
| FUS6 | SK_203944 | TGCACCAAAAAGCAGAAGATT | TGTCCTTAAGATCCATGAAACC | F+p754 or R+p745 |
| | SK_062171 | TGTAAATTTGGCAGCAAACAA | TCATAGCTTCGCGTACAGACA | R+p745 |
| FUS6 | WiscDsLox-453-456F5 | TACGGCTACCCATGATAAACC | TTGCATGGTAATTGAAATGAGAA | F+p745 |
| | WiscDsLox-402F03 | TGCATTTCTTTGAAGTTGTTCTC | TTTGTTGAGCGTTTCATTTGTT | F+p754 or R+p745 |
| FUS7 | SK_073580 | GAATAGCCACAAGGAGGAACC | TTGGAGCATTTATGAGATCGAG | F+p754 or R+p745 |
| | SK_150313 | TCTCAGACGCGTATTCTCTCC | TTTCCGGATGACGAATCTGTA | F+p754 or R+p745 |
| FUS11 | SK_063592 | ATGGCAACAGATCCGAGAAA | CGGTTTAGCCAGACCAATTT | F+p754 or R+p745 |
| | SK_123180 | GCGTCTCTCTTCCAATTCGT | CCCGTAATCCCACCCTTATC | F+p754 or R+p745 |

**Notes.**
[a]"T-DNA PCR indicates" which gene-specific primers can be used to detect the T-DNA insertion when used in combination with the T-DNA left border primer p745.

## RESULTS

### Design of the genetic screen

We use the term "deletion caused by T-DNA recombination" to refer to the situation where meiotic crossing over between T-DNA insertions located nearby to each other leads to deletion of the intervening genomic DNA (Fig. 1). In our previous study describing the identification of a deletion caused by T-DNA recombination in the *MEKK1/2/3* gene family, we used a PCR-based method to identify a single deletion mutant after screening ca. 2,000 seedlings (*Su et al., 2013*). In the present study, our objective was to screen a larger population of individuals in order to better ascertain the frequency with which deletions caused by T-DNA recombination occur. In order to do this in a cost-effective manner, we chose to develop a genetic screen that would let us visually screen for evidence of deletions caused by T-DNA recombination. The rationale for the screen was to choose loci in the *Arabidopsis* genome for which null mutants produced a striking visual phenotype that could be easily scored at the early seedling stage. We chose to use *FUSCA* (*FUS*) genes for this purpose (*Misera et al., 1994*). The *FUS* genes are negative regulators of photomorphogenesis such that null mutants cause dark-grown seedlings to display characteristics normally seen in light-grown seedlings. When *Arabidopsis* seeds are germinated and grown in the dark for four days, the hypocotyls elongate, the apical hook remains closed, and the seedlings are pale white in appearance. By contrast, *fus* mutant seedlings grown in the dark have short hypocotyls, the apical hook is open, and they are dark purple due to anthocyanin accumulation (*Misera et al., 1994*). A *fus* mutant can therefore be easily distinguished on a plate of thousands of wild-type dark grown seedlings as a short purple seedling with open cotyledons against a background of long, skinny, white seedlings. This made the *fus* mutant phenotype a good candidate for our genetic screen. There are at least fourteen *FUS* loci (*Misera et al., 1994*) in the genome, and these genes have also been identified as *CONSTITUTIVE PHOTOMORPHOGENIC* (*COP*) and *DE-ETIOALTED* (*DET*) (*Huang, Ouyang & Deng, 2014*).

**A.**

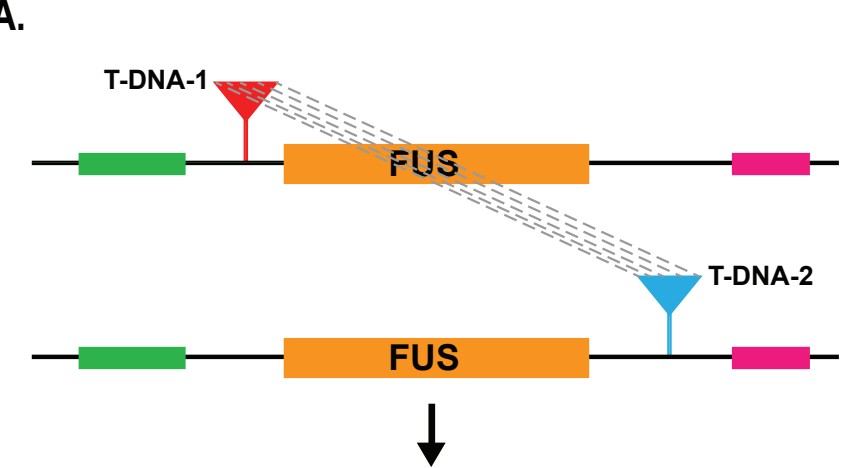

**Crossing over between T-DNA-1 and T-DNA-2 leads to deletion of chromsomal DNA on one of the recombined chromosomes**

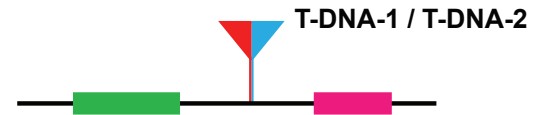

**B.**

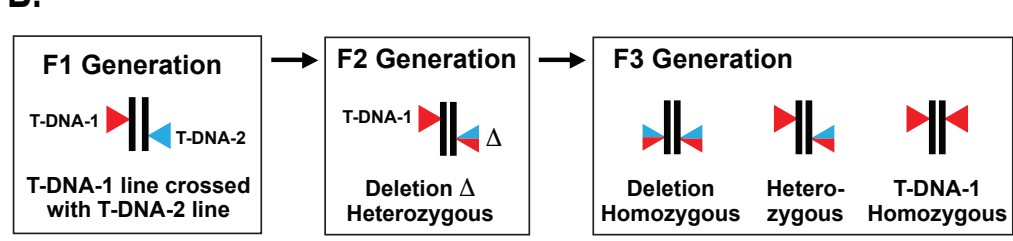

**Figure 1** **Schematic of T-DNA directed deletion and genetic screen.** (A) Recombination between the T-DNA insertions results in deletion of the *fus* gene. (B) A T-DNA directed deletion that occurs during meiosis in an F1 parent will not produce a visible *fus* phenotype until the F3 generation.

To identify specific *FUS* loci for our screen, we searched the available T-DNA mutant collections for *FUS* genes that had T-DNA insertions in close proximity upstream and downstream of the *FUS* gene. Suitable T-DNA lines were found for *FUS2*, *FUS6*, *FUS7*, and *FUS11* (Fig. 2). In the case of *FUS6*, we chose two different sets of T-DNA lines flanking *FUS6*, one set from the SALK T-DNA line collection (*Alonso et al., 2003*) and the other from the WiscDsLox collection (*Woody et al., 2007*). The T-DNA lines for *FUS2*, *FUS7*, and *FUS11* were from the SALK collection.

In our previous study involving deletion of the *MEKK1/2/3* locus, we determined that T-DNA left border sequences were present on both sides of the T-DNA inserts for *MEKK1* and *MEKK3*. This observation suggested that the at least two copies of the T-DNA

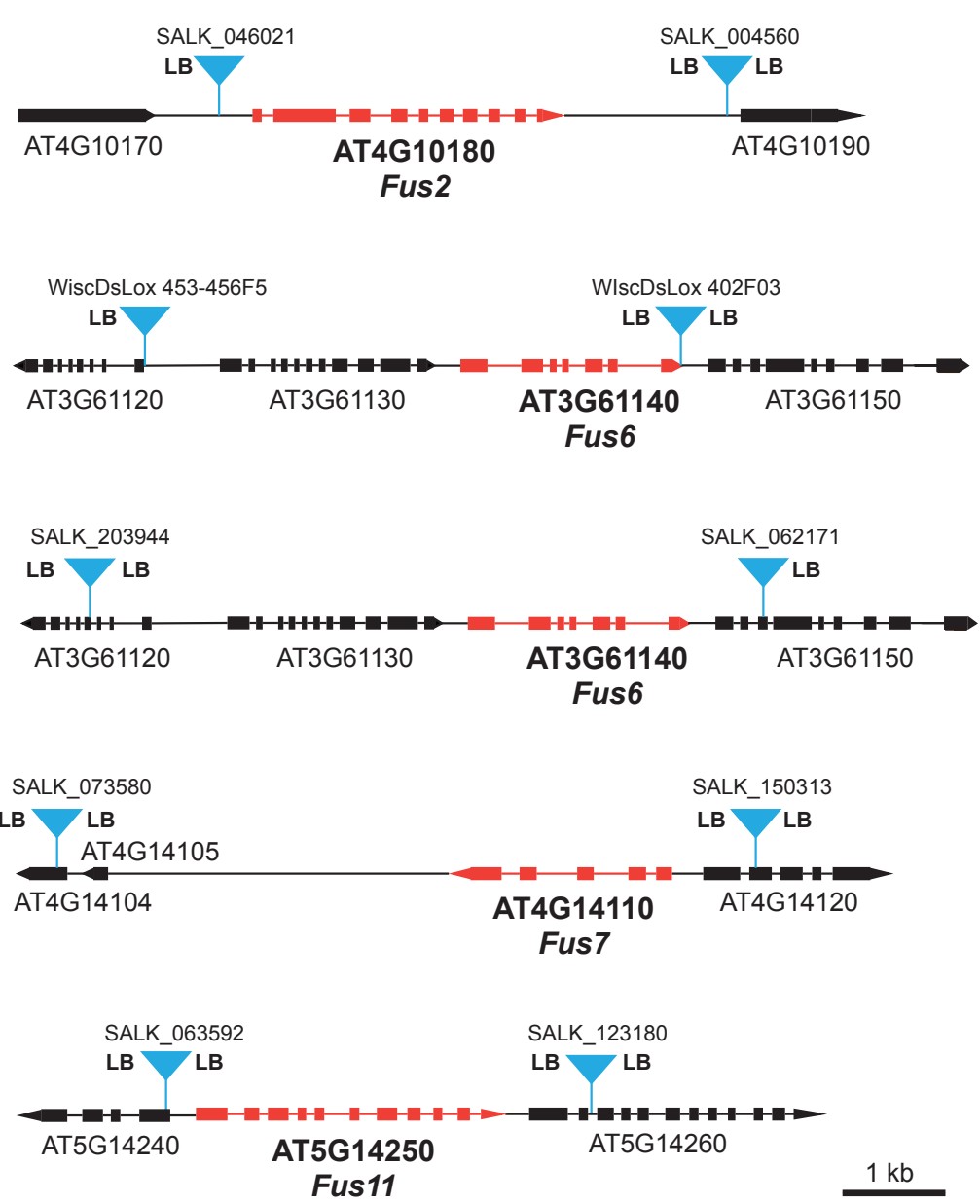

**Figure 2  Maps of *FUS* loci used in this genetic screen.** Positions of the T-DNA inserts used for each *FUS* locus are indicated with blue triangles. The presence of the T-DNA left border sequence is indicated by "LB". "LB" on both sides of the T-DNA suggests that there are at least two copies of the T-DNA element inserted in an inverted repeat orientation. All maps are drawn to the same scale as indicated by the scale bar.

monomer were inserted at each locus in an inverted repeat orientation. Since inverted repeats have the potential to form hairpin structures that may have the ability to stimulate recombination, we were interested in determining if some of the T-DNA inserts used in the present study had left border sequences on both sides of the insert. PCR reactions using left border-specific primers and gene specific primers were used to map the left border

**Table 2** Genetic screen results.

| Gene | T-DNA 1 | T-DNA 2 | # of F2 Parents | # of Deletions |
|------|---------|---------|-----------------|----------------|
| FUS2 | SK_046021 | SK_004560 | 2,916 | 0 |
| FUS6 | SK_203944 | SK_062171 | 2,592 | 0 |
| FUS6 | WiscDsLox 453-456F5 | WiscDsLox 402F03 | 3,312 | 0 |
| FUS7 | SK_073580 | SK_150313 | 2,583 | 0 |
| FUS11 | SK_063592 | SK_123180 | 2,754 | 0 |
| Total | | | 14,157 | 0 |

sequences present at each locus. As shown in Fig. 2, at least one of the T-DNA inserts in each pair had left border sequence on both sides of the insert. In the case of *FUS7* and *FUS11*, both the upstream and downstream T-DNA inserts had left border sequences on both sides.

For a given *FUS* gene, the upstream and downstream T-DNA lines were crossed together to produce F1 plants carrying two T-DNA insertions: one upstream of the *FUS* gene and one downstream (Fig. 1). If recombination occurs between those two T-DNA insertions during meiosis, one would expect one of the meiotic products to carry a deletion of the *FUS* gene. Unless both the pollen and egg carried deletion events, one would expect T-DNA directed deletions produced by an F1 plant to be heterozygous in the F2 progeny (Fig. 1). Since the *fus* phenotype is a recessive condition, it is necessary to perform F3 progeny testing to identify the presence of a T-DNA directed deletion carried by an F2 plant. To streamline this process, we grew nine F2 plants together in a single pot and collected their F3 progeny in bulk. Ca. 1,200 F3 seeds from each pot were then germinated in the dark on moist filter paper and visually scored after four days to screen for the *fus* phenotype. This represents an average of 133 progeny for each F2 parent. If any of the F2 parents were carrying a heterozygous deletion of the *FUS* gene, then we would expect 25% of those progeny, or ca. 33 seedlings, to display the *fus* phenotype. In dark growth conditions, *fus* seedlings are short and purple with open cotyledons, which would stand in stark contrast to the long, white, etiolated form of wild-type seedlings.

### Results of genetic screen

F3 progeny collected from a total of 14,157 F2 parent plants were screened for the *fus* phenotype as described above in order to find seedlings potentially carrying deletions caused by T-DNA recombination of the *FUS* gene (Table 2). Since the recombination event that would lead to a deletion could occur during meiosis in either the male of female gametophyte, the population of 14,157 F2 plants that we screened represented 28,314 meioses. The distribution of these F2 plants between the different *FUS* loci used in the study are shown in Table 2. No *fus* mutant seedlings were observed in any of the F3 progeny tested.

## DISCUSSION

The objective of this study was to determine the rate with which recombination occurs between T-DNA inserts located nearby to each other in the *Arabidopsis* genome. One of the consequences of this type of recombination is the deletion of chromosomal DNA located between the two T-DNA inserts. We use the term "deletions caused by T-DNA recombination" to describe this phenomena. This type of deletion is caused by recombination between two established T-DNA inserts that were generated via independent transformation events in separate plants. Recently, it has been shown that during the initial T-DNA integration process that heterogeneous T-DNA integration can cause chromosomal rearrangements, including deletions, as part of the double-stranded break repair mechanism driving T-DNA insertion (*Hu et al., 2017*). This type of T-DNA associated deletion is mechanistically distinct from the phenomena that we were targeting with our study.

In our previous work we observed one instance of a deletion caused by T-DNA recombination in a population of ca. 2,000 seedlings (*Su et al., 2013*). Because of the extensive collections of *Arabidopsis* T-DNA insertion lines available through public stock centers, we were intrigued by the possibility that deletions caused by T-DNA recombination could provide an effective strategy for genome engineering. If the rate of deletion caused by T-DNA recombination was high enough, one could imagine obtaining a pair of T-DNA lines flanking a genomic region of interest, producing an F1 parent carrying those two T-DNA insertions, and then screening for deletions in the F2 progeny. Since the genetic screen described in this study did not reveal any evidence for deletions caused by T-DNA recombination from a population of 14,157 F2 parent plants, it appears that this type of deletion may be a relatively rare event.

The most straightforward interpretation of our failure to observe *fus* mutant seedlings in our genetic screen is that deletions caused by T-DNA recombination did not occur in the F1 plants used in our study. It is also possible, however, that deletions may have been generated, but were subsequently selected against due to lethality or reduced fitness of the specific deletion allele that was formed. In most of the *FUS* loci used in our study, the upstream and downstream T-DNA inserts are located in neighboring genes. It is possible that deleting those neighboring genes in addition to the *FUS* gene could have a detrimental effect on the plant. It should be noted that each of the T-DNA lines used for this study was viable when the T-DNA insert was homozygous, indicating that the neighboring genes are not essential genes and that deleting them on their own should not cause lethality. It nevertheless remains possible that genetic interactions between the loci or other functional elements contained within the intervals between the flanking T-DNAs could lead to unexpected lethality of the predicted deletions.

## CONCLUSIONS

Our study has suggested that the deletion caused by T-DNA recombination phenomena is unlikely to be a useful strategy for genome engineering in *Arabidopsis*. Given recent advances in the application of CRISPR-Cas9 methods in plants, a more effective approach may be to direct double-strand breaks using Cas9 targeted to sites flanking the region

of interest (*Yin, Gao & Qiu, 2017*). This type of approach has been used in rice to delete regions up to 245 kb in size (*Zhou et al., 2014*) and *Arabidopsis* to delete regions up to 120 kb (*Ordon et al., 2017*). Despite the lack of an immediate practical application, our previous observation of recombination between established T-DNA insertions highlights the dynamic nature of genome structure. As our ability to modify genomes advances, there remains a need to more fully understand all of the mechanisms and processes that shape genome structure and behavior.

### Funding
This work was supported by the National Natural Science Foundation (MCB-1407063). The funders had no role in study design, data collection and analysis, decision to publish, or preparation of the manuscript.

### Grant Disclosures
The following grant information was disclosed by the authors:
National Natural Science Foundation: MCB-1407063.

### Competing Interests
The authors declare there are no competing interests.

### Author Contributions
- John F. Seagrist performed the experiments, analyzed the data, contributed reagents/materials/analysis tools, prepared figures and/or tables, approved the final draft.
- Shih-Heng Su conceived and designed the experiments, performed the experiments, analyzed the data, contributed reagents/materials/analysis tools, approved the final draft.
- Patrick J. Krysan conceived and designed the experiments, analyzed the data, prepared figures and/or tables, authored or reviewed drafts of the paper, approved the final draft.

### Data Availability
The research in this article did not generate any data or code.

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
