# Peer review of "Recombination between T-DNA insertions to cause chromosomal deletions in Arabidopsis is a rare phenomenon"

_PeerJ, doi:10.7717/peerj.5076_

## Round 0.1 · original submission · Minor Revisions

· Academic Editor

Minor Revisions

With the recent findings and application of gene editing techniques the impact of this work may have lost some of its relevance; however, your perseverance to test for potential deletions based on a previous observation serve to highlight that an experiment was done, though with a negative outcome. The reviewer comments were not too critical sufficing that minor modifications be addressed in a revision of the manuscript. In light of today's technologies, perhaps some reflection on your part on how such studies still have value in understanding the nature of genome structure and behaviour may add value, as we are now entering the age of pick-and-choose genome manipulation. It may be easier, but do we really understand enough at this point? Thank you for the contribution.

Reviewer 1 ·

Basic reporting

This paper is written clearly and easy to follow.

Experimental design

No comments

Validity of the findings

Technologically, genome deletion is significant not only for producing multiple mutants but also for functional analysis of non-coding sequence. Inspired by the authors’ early discovery about MEKK1, they designed a strategy based on recombination between neighboring T-DNA to delete genome sequence. Deleting genome with high efficiency, even at the age of CRISP/Cas9, is still challenging. Although the result is negative, the efforts are helpful for us to understand T-DNA recombination.

Additional comments

The major points are:
1. The term of “T-DNA directed chromosomal deletion” may be misleading. Recently a report showed heterogeneous T-DNA integration can result in major chromosomal rearrangements including large deletion (Hu et al., 2017). Accordingly, the so called “T-DNA directed deletion” in this work may occur at the stage of T-DNA integration or later stage (through recombination between T-DNA). I would suggest the authors cite that study and emphasize recombination process in the term.
2. Generally, crossover between two homologous chromosomes at a specific site is rare. The finding about recombination between MEKK1 and MEKK3 may result from their unique T-DNA structures which may increase the recombination rate. In the original report about MEKK1 (Su et al., 2013), the authors suggested multiple copies of T-DNA were inserted head-to-head with left border at both sides. Whether features like multiple-copy insertion and orientation would influence the rate? This work did not tell us any structural features of T-DNA insertion sites.

Reviewer 2 ·

Basic reporting

The article is concisely written. The Introduction and Materials & Methods sections are well-presented and contain necessary information. The references appear appropriate and up-to-date. The structure conforms to the PeerJ standards. The figures, which all appear to be vector graphics, are of sufficient quality, well-labeled and also sufficiently described, as they are basically self-explaining. The figures are relevant for the understanding of the manuscript and are thus of importance, but the main findings are summarized in Table 1, rather. The inclusion of raw data is not really applicable to the current article, as the raw data is presented in table 1 as numbers, and there is no original data beyond that.

Experimental design

The current submission has a very simple question at its basis: Can T-DNA insertions, present in two different lines, be exploited for the creation of deletions via recombination between the homologous sequences (the T-DNAs) upon crossing of the respective lines? This had been observed in a previous report (Su et al 2013), where a deletion of ~ 25 kb had been detected upon crossing two T-DNA insertion lines.
In the current study, T-DNA insertion lines with insertions flanking FUS genes were crossed to provide a visual output of such recombination events, by elimination of the respective FUS gene(s), visible by photomorphogenesis in dark-grown seedlings (vs etiolated seedlings). The phenotype is well-chosen, as fus mutants can be easily identified among wild type, etiolated seedlings, and also the repertoire of FUS genes allows testing of different genes/T-DNA insertions in a unified experimental design.
The underlying questions is of interest, as the proposed technique might represent a valid approach to eliminate clustered genes and/or gene families, and fits the aims and scopes of the journal. The research question is very simple, as stated above (can the proposed approach be used to create deletions?), and well-defined. The analysis appears solid and the methods are well-described.

Validity of the findings

The findings are presented in a clear manner, although the outcome surely is somewhat disappointing for the authors: The tested approach does not work; the previously observed chromosomal deletion was a rare event picked up by chance, apparently. The presented data appears robust, as several different crosses / T-DNA insertion lines were tested. Statistics are not really relevant, as no recombination events were detected. The conclusion that such recombination events are rare is clearly stated and answers the research question: This approach is not very promising. Alternative strategies for obtaining such deletions via sequence-specific nucleases (CRISPR/Cas) are indicated in the conclusions section, which provides guidance for those intending to generate deletions.

Additional comments

Overall, the article is well-written, and provides a clear answer to the previously stated hypothesis that adjacent T-DNA insertions might give rise to chromosomal deletions upon crossing of respective lines. Considering that the previous statement that such phenomena could be exploited for genetic engineering yet remained unanswered, it is important to have this information available in the public domain. However, a few aspects should be clarified and/or corrected:

- gene names should be consistently written in italics, also e.g. Arabidopsis thaliana or Agrobacterium tumefaciens as species names should be in italics

- there are a few typos, as for example CONTITUTIVE PHOTOMORPHOGENIC in line 164 (constitutive),

- as far as I am aware (although I cannot provide a suitable citation here), inverted repeats (T-DNA insertions in opposing senses) rather than direct repeats (T-DNA insertions in identical orientations) may give rise to the "desired" recombination events. I think that it would therefore be important to include this info in the text: What was the orientation of T-DNAs in the previous report (Su et al 2013), and which orientations were tested in the genetic screen presented here? For Su et el., this can be mentioned in the text, and indicated for combinations tested here in figure 2.

---

## Round 0.2 · accepted · Accept

· Academic Editor

Accept

Thank you for your considerations addressed based on the reviewers comments. The potential for identifying a recombination event was well explained, with choice of a reasonable detection system; the numbers screened also appeared sufficient to claim rarity in occurrence, and adequate explanation for why recombination was not observed. The notion to address the hypothesis is commendable and worth noting; thus, the manuscript is considered accepted for publication. There may be a few places to address other minor corrections; however, the manuscript appears in well enough shape to be moved forward. Congratulations.

Line 36: “in trans” should be in italics.
Line 80: I think the “kinase kinase kinase” is intentional, and can be read that way; it just throws off the usual reader, or emphasizes that we are dealing with repeats, but only mentions the MAPKK version instead of the MAPKKK.
Line 136: “Petri” can be capitalized.
Line 211: “male or female” instead of “male of female”.
Line 255: “CONCLUSIONS” instead of “CONSLUSIONS”.

#